# Exposure to a Standardized Catastrophic Scenario in Virtual Reality or a Personalized Scenario in Imagination for Generalized Anxiety Disorder

**DOI:** 10.3390/jcm8030309

**Published:** 2019-03-05

**Authors:** Tanya Guitard, Stéphane Bouchard, Claude Bélanger, Maxine Berthiaume

**Affiliations:** 1Département de Psychologie, Université du Québec à Montréal, Montréal, QC H3C 3P8, Canada; tanya.guitard@videotron.ca (T.G.); belanger.claude@uqam.ca (C.B.); 2Département de Psychoéducation et de Psychologie, Université du Québec en Outaouais, Gatineau, QC J8X 3X7, Canada; 3School of Psychology, University of Ottawa, Ottawa, ON K1N 6N5, Canada; mbert094@uottawa.ca

**Keywords:** Generalized Anxiety Disorder (GAD), virtual reality, exposure in virtual reality, cognitive exposure, standardized scenario, personalized scenario

## Abstract

The cognitive behavioral treatment of generalized anxiety disorder (GAD) often involves exposing patients to a catastrophic scenario depicting their most feared worry. The aim of this study was to examine whether a standardized scenario recreated in virtual reality (VR) would elicit anxiety and negative affect and how it compared to the traditional method of imagining a personalized catastrophic scenario. A sample of 28 participants were first exposed to a neutral non-catastrophic scenario and then to a personalized scenario in imagination or a standardized virtual scenario presented in a counterbalanced order. The participants completed questionnaires before and after each immersion. The results suggest that the standardized virtual scenario induced significant anxiety. No difference was found when comparing exposure to the standardized scenario in VR and exposure to the personalized scenario in imagination. These findings were specific to anxiety and not to the broader measure of negative affect. Individual differences in susceptibility to feel present in VR was a significant predictor of increase in anxiety and negative affect. Future research could use these scenarios to conduct a randomized control trial to test the efficacy and cost/benefits of using VR in the treatment of GAD.

## 1. Introduction

In the treatment of anxiety disorders, exposure is defined as “any procedure that confronts the person with a stimulus which typically elicits an undesirable behavior or an unwanted emotional response” [1] (p. 121). This stimulus can take an animate form (animal, insect), inanimate (heights, storms), a situation (public speaking), or even a thought (memories of a traumatic event, anticipation of a catastrophe). Regardless of the stimulus, the purpose of exposure is to learn new mental associations between the stimuli and lack of threat [2,3]. Considering that exposure requires confronting feared stimuli, the confrontation is associated with an increase in anxiety [4]. Studies have demonstrated that, compared to people who do not suffer from an anxiety disorder, immersions in virtual reality (VR) can elicit anxiety in people suffering from specific phobia [5], and from obsessive-compulsive disorder characterized by fear of contamination [6] or by checking behaviors [7]. Several studies, summarized in literature reviews (e.g., [8]) and meta-analyses (e.g., [9,10,11]), have documented the relevance and efficacy of using VR to conduct exposure (also called *in virtuo* exposure, [12]) in the treatment of anxiety disorders. 

VR is defined as the product of using computer and behavioral interfaces to simulate the behavior of 3D entities that interact in real time with each other and with a user immersed via sensorimotor channels [13]. VR systems are designed so images viewed in a head mounted display (HMD) change according to the user’s head movements. When immersed in VR, individuals can explore different environments, allowing them to feel as if they are physically in this synthetic environment [14]. This feeling of being “in” the virtual environment is called ‘’presence’’ [15] and is considered an important feature of VR. 

Conducting exposure *in virtuo* in the context of cognitive behavior therapy (CBT) has several benefits when compared to in vivo exposure. First, *in virtuo* exposure allows a greater control over the environment, which can be practical for both therapists and patients [16]. Unanticipated events (e.g., poor conditions during an airplane flight or unexpected animal/insect reactions) are less likely, allowing a more controlled exposure session. *In virtuo* exposure also provides greater standardization of the exposure cues, which can be useful for researchers as well as clinicians. Finally, using VR is considered more attractive than in vivo exposure for some patients [17]. 

However, creating virtual environments for *in virtuo* exposure for the treatment of generalized anxiety disorder (GAD) could be more complex than with other anxiety disorders. GAD is defined by excessive anxiety or worries almost every day for at least six months and concerning a variety of themes. It is characterized by fear of negative and uncertain future events [18,19,20]. As opposed to other anxiety disorders, people suffering from GAD are not essentially afraid of specific and concrete stimuli. They are afraid of uncertainty accompanied with a broad range of situations. In the cognitive-behavioral treatment of GAD, exposure is often conducted in the imagination, where patients have to repeat mentally or write down a scenario depicting one of their worst catastrophic worries [21,22].

Using VR for exposure with GAD has its share of advantages [23]. For example, not all patients are good at imagining or visualizing feared situations, yet exposure in imagination is often used with GAD patients. It may be difficult to know what patients are thinking about, if they are engaged in avoidance or neutralization behaviors while doing exposure, or if the right stimuli are included in the scenario. With respect to neutralization, subtle avoidance and safety seeking behaviors, using standardized scenarios reduces the risks of avoidance because the content of the scenario is known and visible to the therapist during the immersion in VR. 

Previous work has been conducted to identify common ingredients in the exposure scenarios of actual GAD patients [24] and to proposed standardized scenarios that can be used for exposure [25]. Empirical data collected with GAD patients exposed in imagination to their personal worry scenarios or to the standardized ones showed that standardized scenarios can elicit anxiety (as measured with self-report and heart rate) and negative affect [25]. In their research, Guitard and her colleagues [25] demonstrated that the effect size of exposure in imagination to the individualized scenarios was slightly higher than the standardized ones on the measure of heart rate but, nevertheless, the results were positive enough to warrant transposing the scenarios used in imagination into VR scenarios and testing them with people suffering from GAD. 

Accordingly, the goal of the current study is to document the potential of virtual environments adapted from catastrophic scenarios to induce the anxiety necessary to eventually use VR as an exposure strategy in the treatment of GAD. Three exposure scenarios are compared: (a) exposure to a neutral virtual environment; (b) exposure in imagination to a personalized scenario (IM-Exp); and (3) exposure in VR to a standardized scenario (VR-Exp). Each participant was exposed to all three conditions, first to the neutral environment and then to exposure in imagination or in VR, in random order. The hypothesis was that exposure in VR will induce more anxiety than the neutral scenario. Power estimations were performed before the study, with medium to large effect sizes expected for this hyposhesis based on results from other studies [5,6,25]. To prevent having to prove the null hypothesis (i.e., *in virtuo* being as effective as in vivo) without justification, no formal hypothesis was expressed for the comparison between exposure in VR and in imagination, and no power estimation was conducted a priori.

## 2. Materials and Methods

### 2.1. Participants

Inclusion criteria for this study were as follows: (a) primary diagnosis of GAD, (b) aged between 18 and 65, and (c) never having experienced VR before. The Randot Stereo™ test from Western Ophtalmics was used to assess if participants have stereoscopic vision. Exclusion criteria consisted of: (a) suffering from an anxiety disorder other than GAD as primary diagnosis; (b) suffering from claustrophobia, because the experimentation was held in an immersive CAVE-Like system, a rather confined area; (c) use of benzodiazepines, because of the impact this type of medication might have on the variables measured; and (d) suffering from any of the following health issues (due to potential interactions with VR): diseases related to the inner ear or vestibular system, cardiovascular diseases or circulatory disorders, migraines, blood pressure disorders or diabetes. The final sample included a total of 28 participants who all met the criteria.

### 2.2. Procedure

Participants were recruited through the Université du Québec en Outaouais (UQO) via email and posters. The project, conducted in concordance with the Declaration of Helsinki and the Canadian Tri-Council policy statement for ethical conduct for research involving humans, was approved by the Ethics Committee of UQO and participants signed a free and informed consent form. All individuals who wanted to participate in the study were first briefly screened by phone in order to assess whether or not excessive worry seemed present. In cases where anxiety was caused by another disorder, they were referred to another service. When GAD seemed probable and exclusion criteria were screened, a first meeting was scheduled to proceed to the complete evaluation using the ADIS-IV. The participants included in the study were randomly assigned to one of the two conditions: (a) exposure to a standardized scenario in VR followed by exposure to an individualized scenario in imagination (VR-Exp/IM-Exp; *n* = 13) or (b) exposure to an individualized scenario in imagination followed by exposure to standardized scenario in VR (IM-Exp/VR-Exp.; *n* = 15). At the end of the assessment session, all participants were given a battery of questionnaires to complete alone at home (without consulting other people) and return at the following meeting. In session 2, participants were asked to identify a worry theme and write a catastrophic scenario based on their worst fear. The writing of the scenario began during the session to allow time for the therapist to review the content and give feedback to the participant. Following this, participants were asked to further improve their scenario, if needed. They were told not to read their scenario at home to avoid habituation. A third and final session was scheduled in which the exposures took place. Each participant was first immersed in a neutral non-catastrophic virtual environment for 5 min that consisted of a quiet and empty room with a glass door and the sun shining in from large windows. Participants were asked to physically walk around the room to familiarize themselves with VR. Following the first experimental exposure scenario (either in VR or in imagination), a distraction task was performed where participants were asked to dash the A letter each time it occurred in a random and incomprehensible text. Following the distraction task, the other experimental exposure session took place. At the end of the third session, debriefing was completed to ensure the well-being of the participants following the brief exposure to the anxiety provoking scenarios and offer relevant clinical referrals, if necessary. 

### 2.3. Virtual Environments (VE)

The three standardized exposure scenarios used in imagination by Guitard et al. [25] were recreated in VR and the experimenter selected one based on the main worry theme as reported by the participant: (a) VE 1: an emergency room (used with 11 participants), (b) VE 2: an apartment (used with 15 participants), or (c) VE 3: a student room (used with 2 participants). 

#### 2.3.1. VE 1: Emergency Room

This environment was created with the intention of eliciting worry in participants suffering mostly from worries related to health. The participant was immersed in a hospital emergency waiting room. Other patients are nearby and display symptoms of sickness (coughing) or fatigue. One is wearing a disposable face mask. Sounds are heard, such as a mother crying after receiving bad news concerning her child, and a wife is told that nothing can be done to save her husband, etc. Other patients are called to see the doctor, but not the participant. At some point, doctors come into the waiting room and discuss a case while looking at the participant, who cannot, however, understand what they are saying. 

#### 2.3.2. VE 2: Apartment

The setting for this scenario is an apartment that participants are invited to visit. Participants first overhear a conversation taking place behind closed doors suggesting that an accident has occurred. Afterwards, a special announcement is made on the radio regarding recent burglaries in the neighborhood. At the same time, a rock is thrown at the window, and when the participant looks out the window, a group of men is seen roughing up another individual. Finally, a brief message is left on the answering machine. Seven message options are available to choose from: (a) the police calling because a loved one was involved in an accident, (b) the doctor calling regarding test results that were previously overlooked, (c) the participant’s spouse, either male or female, is saying that they have met a new lover and are leaving, (d) the bank needs to be called back regarding several late payments and is threatening to take action, (e) a receptionist from work calling regarding recent budget cuts and a problem involving the participant, (f) the university calling regarding unpaid tuition fees and the impossibility of registering for the semester, or (g) the school is calling regarding the participant’s child’s recent behavior and academic problems.

#### 2.3.3. VE 3: Student Room

The third environment, created for students, alludes to both academic difficulties and social isolation. The participant is in his or her room having to study for upcoming exams. Scattered unpaid bills are visible and suggest financial difficulties. Voices are heard coming from the hall, talking about a student who is failing out of the program. Roommates heard from another room are planning a party to which the participant is not invited and that might, furthermore, disrupt his or her study time.

### 2.4. Measures

#### 2.4.1. Diagnostic and Clinical Measures of Severity

An initial diagnosis of GAD was made using the ADIS-IV at the first session. Questionnaires were then given to each participant to be filled out at home and brought back at the next session. Those questionnaires were used to further assess each participant and assess the clinical severity of the sample as well as describe VR factors that may influence the results.

##### Diagnostic Measure: Anxiety Disorders Interview Schedule for DSM-IV (ADIS-IV)

This semi-structured interview allows for a thorough evaluation of anxiety disorders as well as mood disorders, substance-abuse disorders, and somatoform disorders as they hold the higher comorbidity rates with anxiety disorders. The ADIS-IV [26] was used for diagnostic purposes and the severity score on the diagnosis of GAD is reported to describe the sample.

##### Clinical Measure: Penn State Worry Questionnaire (PSWQ)

The French translation of this questionnaire [27] comes from Ladouceur et al. [28]. This 16-item questionnaire measures the level of worry typical to GAD on a scale of 1 to 5. The psychometric properties of the English version are very good, with good internal consistency (Cronbach’s alpha ranging from 0.86 to 0.95) and test retest reliability (ranging from 0.74 to 0.93) [29,30]. The same was found for the French version (see [30]) indicate equally good validity and internal consistency. The PSWQ was used to describe the sample.

##### Clinical Measure: Cognitive Avoidance Questionnaire (CAQ; Original French Version)

Cognitive avoidance plays an important role in maintaining excessive worry in GAD. This measure [31] was therefore used to evaluate the degree of cognitive avoidance in the sample. Studies have indicated good psychometric features for this scale with a Cronbach’s alpha of 0.95 for the totality of the items on an adult sample and of 0.92 on a sample of adolescents, both non-clinical samples.

#### 2.4.2. Measures of Users’ Experience in Virtual Reality 

These questionnaires measure important concepts in clinical applications of VR. They allow comparing reactions of participants from one study to another.

##### Users’ Experience: Presence Questionnaire (PQ)

The PQ is a French-Canadian translation (validated by the Cyberpsychology Laboratory of UQO [32]) of the Witmer and Singer Presence Questionnaire [33]. This questionnaire contains 24 items in the form of closed-ended questions, on a scale of 1 (“not at all”) to 7 (“completely”), and as a measure of presence it is useful to describe how participants perceive the properties of the virtual environments and the technology used. Cronbach’s alpha reaches 0.84. The duration of administration was approximately 7 min. 

##### Users’ Experience: Gatineau Presence Questionnaire (GPQ)

This questionnaire was created by the Cyberpsychology Laboratory as a brief supplement to the PQ to address the experience felt by the users while immersed [6]. It includes four questions, on a scale of 0–100. The GPQ has a Cronbach’s alpha of 0.69. 

##### Users’ Experience: Simulator Sickness Questionnaire (SSQ)

This questionnaire is a French-Canadian translation (validated by the Cyberpsychology Laboratory of UQO [34]) of the Simulator Sickness Questionnaire [35] designed to measure the level of unwanted negative effects induced by the immersions in VR. It consists of 16 items, rated on a four-point scale. This questionnaire was administered to participants for the first time at the beginning of the meeting involving virtual reality, in order to know their physical state well before the first immersion (results not shown) and after the immersion in VR. Cronbach’s alpha reaches 0.87. The SSQ was scored according to guidelines from Bouchard et al. [34] and the raw total score is reported.

#### 2.4.3. Dependent Variables

The following questionnaires were used as dependent variables to assess the level of anxiety and negative affect throughout the experimentation process of the third session. Participants had to fill out these questionnaires after each exposure session.

##### Dependent Variable: State Scale of the State Trait Anxiety Inventory—Form Y1 (STAI-Y1)

The French version of this questionnaire [36,37] was used. Only the Y-1 version (state form) was retained for the present study because the goal of the project was to assess anxiety levels at different times during the experimentation rather than evaluate anxiety traits in the participants. The French version of this measure has excellent psychometric values, with a Cronbach’s alpha of 0.94 and 0.86 for men and women, respectively [36]. 

##### Dependent Variable: Negative Affect Scale of the Positive and Negative Affect Schedule (PANAS)

The French-Canadian version [38] of the questionnaire developed by Watson, Clark and Tellegen [39] was used. It consists of two scales; one measuring positive affect and the other negative affect. Only the negative affect subscale is reported here, given the study’s focus on anxiety. Items represent different feelings and emotions that are rated on a 5-point scale ranging from 1 (“very slightly or not at all”) to 5 (“extremely”). Internal consistency of the negative affect subscale is adequate, with a Cronbach’s alpha ranging from 0.80 to 0.84.

#### 2.4.4. Predictors of Levels of Anxiety and Negative Affect During the Immersion in VR

##### Predictor: Intolerance of Uncertainty Scale (IUS; Original French Version)

This questionnaire [20] rates the degree of intolerance to uncertainty using 27 items that describe uncertainty as negative and something to be avoided. Participants have to rate each item on a 5-point Likert scale ranging from 1 (“not at all characteristic of me”) to 5 (“entirely characteristic of me”). This measure possesses very good internal consistency (Cronbach’s alpha of 0.91) and good convergent validity with the PSWQ. Also, the main advantage of this measure is its good sensibility and specificity to excessive worry that allows it to be administered more than once during treatment to assess progress [21]. The IUS, and the following two questionnaires, were used to describe the sample and explore potential predictors of patient’s reactions in VR.

##### Predictor: Why Worry-II (WW-II)

This questionnaire is a revised version of the original Why Worry [31]. It assesses positive beliefs about worry with five different subscales: (1) worry as a problem solving tool; (2) worry helps motivate; (3) worrying protects and prepares in the face of a negative outcome; (4) worrying can, in itself, prevent a negative outcome and (5) worry is a positive personality trait. This self-reported measure contains 25 items ranging from 1 (“not at all true”) to 5 (“absolutely true”). This questionnaire possesses good test-retest reliability (*r* = 0.81) and internal consistency (Cronbach’s alpha of 0.93) [40]. 

##### Predictor: Immersive Tendencies Questionnaire (ITQ)

This questionnaire is a French-Canadian translation (validated by the Cyberpsychology Laboratory of UQO [32]) of the Immersion Tendencies Questionnaire [33] and contains 18 items calculated on a scale of 1 (“never”) to 7 (“often”). This questionnaire aims to measure the predisposition of the individual to experience presence. It was administered only once. Cronbach’s alpha reaches 0.78.

### 2.5. Experimenters and Hardware

Four experimenters, all doctorate students with training in CBT for anxiety disorders, conducted the study. Supervision was made available to them and provided by a licensed psychologist. The immersions in VR were conducted in a 6-side CAVE-Like system using retro projected stereoscopic displays and wireless motion tracking (see Laforest et al. [6] for a technical description and a picture). 

## 3. Results

### 3.1. Sample Description

The sample (*N* = 28) consisted of 24 women and 4 men with a primary diagnosis of GAD. They were all francophone with a mean age of 38.33 (SD = 12.78). According to the PSWQ, participants scored within the range of adults suffering from GAD. Comorbid disorders were diagnosed in 64.3% of the sample—social anxiety being the most frequent (*n* = 8) while others were specific phobias (*n* = 4), panic disorder (*n* = 3), obsessive-compulsive disorder (*n* = 1), and other diagnosis (*n* = 2). Further description of the sample is provided in Table 1. No differences were found between the two conditions on the GAD severity (as assessed with the ADIS-IV and the PSWQ), on how they perceived the quality of the VR system (PQ), and in unwanted negative side effects induced by the immersion in VR (SSQ).

### 3.2. Statistical Analyses

Prior to analyses, all variables to be used in further analyses were examined for accuracy of data entry, missing values, normality of distribution, and univariate outliers. After ensuring that there were no errors in data entry or missing values, we screened for extreme kurtosis and skewness values (below 1.5 or above −1.5), which would indicate non-normal distributions. The negative affect scale of the PANAS (neutral scenario and second exposure) had extreme kurtosis values. We also screened for univariate outliers on the state anxiety scale of the STAI and the negative affect scale of the PANAS. When univariate outliers were found, they were winsorized to the next most extreme but acceptable value in that condition (with a z-score less than 1.96 or above −1.96). To do so, z-scores were first obtained for all variables to be used in further analyses. When a z-score was greater than 1.96 or less than −1.96, the next most extreme but acceptable value in the same condition was found and replaced the extreme value that needed to be winsorized. This procedure eliminated all outliers and extreme skewness and kurtosis values. Parametric analyses were then performed, with descriptive results reported in Table 2 (note that results were similar if the data is not corrected for outliers). 

Following data screening, variables were analyzed with repeated measures ANOVAs, followed by a priori orthogonal within-subjects contrasts. Contrasts focused on the impact of the neutral scenario and first exposure and of the first and second exposures on the cognitive exposure group compared to the virtual exposure group. All Mauchly’s (sphericity) values were non-significant, therefore the non-corrected values were used. To control for Type-I error rate, Bonferroni corrections were applied. Controlling with ANCOVAs for the use of three standardized scenarios did not change the interpretation of the results. When results were not significant, the expected number of participants required to detect a significant difference at alpha = 0.05 with a power of 0.80 is reported based on Cohen [41] to illustrate the magnitude of the differences.

Descriptive information and results for all dependent variables following the ANOVAs are reported in Table 2 and Table 3 respectively. For the main effect of Time, a repeated measures ANOVA showed a significant increase in anxiety as measured with the STAI-Y1, when comparing exposure to the neutral VE and exposure to either a catastrophic scenario (traditional personalized scenario or VR scenario, see Figure 1 for illustration). The interaction was non-significant, indicating that exposure to the traditional personalized scenario over time did not elicit more anxiety than exposure to the virtual scenario. The first contrast revealed that the first exposure to either the traditional personalized scenario or the virtual scenario was significantly more anxiety provoking than exposure to the neutral scenario [*t*(26) = 3.82, *p* < 0.001, eta-squared = 0.22, effect size = large, power = 0.96]. The interaction contrast was non-significant [*t*(26) = −0.58, *p* > 0.05, eta-squared = 0.006, effect-size = very small, power = 0.10, expected *N* to detect a significant difference with a power of 0.80 > 2000], showing that both scenarios induced anxiety. The contrast from the first exposure to the second exposure was non-significant [*t*(26) = 0.48, *p* > 0.05. eta-squared −0.004, effect-size = very small, power = 0.07, expected *N* to detect a significant difference with a power of 0.80 > 2000], suggesting that the first exposure was not more anxiety provoking than the second exposure, regardless of the scenario. However, the interaction contrast was significant [*t*(26) = 2.20, *p* < 0.05, partial eta-squared = 0.09, effect-size = medium, power = 0.65], indicating that the traditional personalized scenario elicited more anxiety than the VR scenario. The interaction did not remain significant when applying the Bonferroni correction. 

Results on the negative affect scale of the PANAS were somewhat different (see Figure 2 for illustration). In the group first exposed to traditional cognitive exposure, negative affect decreased in the second exposure whereas the second group, first exposed to VR, shows an increase in negative affect when exposed to the traditional personalized scenario. The results for the main effect of Time from the ANOVA revealed non-significant increase in negative affect overall. The first a priori contrast indicated that negative affect did significantly increase from the neutral scenario to the first exposure in both scenarios [*t*(26) = 2.59, *p* < 0.05, eta-squared = 0.11, effect-size = medium, power = 0.78], although the increase did not remain significant when applying the Bonferroni correction. The interaction contrast was non-significant [*t*(26) = 0.94, *p* > 0.05, eta-squared = 0.02, effect-size = small, power = 0.20, expected *N* to detect a significant difference with a power of 0.80 = 344], revealing a similar and only slight increase in negative affect. The second a priori interaction contrast was non-significant [*t*(26) = 2.03, *p* > 0.05 eta-squared = 0.07, effect-size = medium, power = 0.60, expected *N* to detect a significant difference with a power of 0.80 = 120], although the effect size was close to significance.

Further exploratory analyses were conducted to study predictors of the impact of the exposure to the standardized catastrophic scenarios in VR. To respect a subject-to-variable ratio that minimizes parameter inflation and maximizes replicability, only three predictors were selected: two variables related to GAD (intolerance of uncertainty—IUS, and beliefs about worry—WW-II) and one related to VR (immersive tendencies - ITQ). Prior to performing the analyses, data were screened for linearity (by examining a scatterplot), multicollinearity (by verifying the tolerance and VIF values), autocorrelation among the residuals (by verifying the Durbin-Watson values and examining a scatterplot), multivariate normality (by examining a histogram), and homoscedasticity (by examining a scatterplot). All assumptions were met.

In the exploratory predictor analyses of state anxiety during exposure to the standardized scenario in VR, the main regression was significant [adj*R^2^* = 0.44, *F*_(3, 24)_ = 7.29, *p* < 0.01]. Two predictors were significant, the usefulness of worrying (WW-II; *t* = 2.99, partial *r* = 0.46, *p* < 0.01) and the immersive tendency (ITQ; *t* = 4.14, partial *r* = 0.63, *p* < 0.001). Intolerance of uncertainty was not a significant predictor (IUS; *t* = −0.9, partial *r* = −0.15, *p* = 0.35 ns). The exploratory predictor analyses of negative affect was significant [adj*R^2^* = 0.32, *F*_(3, 24)_ = 4.79, *p* < 0.05], with only the immersive tendency standing out as a significant predictor (ITQ; *t* = 3.54, partial *r* = 0.6, *p* < 0.01). Regression parameters were not significant for the WW-II (*t* = 1.62, partial *r* = 0.27, *p* = 0.12 ns) and the IUS (*t* = −0.22, partial *r* = −0.04, *p* = 0.83 ns). The scaterplots in Figure 3 illustrate the tendecy, and individial differences, for higher predispositions to feel present in VR to be associated with more anxiety and negative affect. 

## 4. Discussion

The goal of the current study was to assess the potential of VR scenarios to elicit anxiety in GAD patients, with the long-term research goal of facilitating cognitive exposure in CBT. The current study compared a standardized scenario in VR and a traditional personalized scenario. More precisely, we compared traditional exposure in imagination using a personalized catastrophic scenario to exposure in VR to a standardized scenario. Exposure to a neutral scenario was used as a baseline for comparisons. We hypothesized that exposure in VR to the standardized scenario would be significantly more anxiety provoking than the neutral scenario. No specific hypothesis was formulated for the comparison between the modalities of exposure. 

Our first hypothesis was supported. The state anxiety scores during exposure in VR and in imagination were significantly higher than the baseline. Results were also in the same range than in the Guitard et al. [25] study, where participants had to imagine the scenarios instead of being exposed to them in VR, and to studies using VR for other anxiety disorders (e.g., [6,7]). The actual difference between exposure to standardized scenarios in VR and personalized scenarios in imagination was significant only when the sequence of exposure sessions was counterbalanced and it did not remain significant after controlling for the number of comparisons. The effect size and statistical power of the comparisons between the two exposure modalities deserve attention. When compared to the neutral scenario, the increase in anxiety experience in the personalized scenario in imagination versus the standardized scenario in VR is associated with a small effect size and more than 2 000 participants would be required to detect a significant difference in the two exposure modalities. This is supporting the potential of VR with GAD patients. However, the direct comparison of the two modalities with each other (i.e., the interaction contrast between the exposure scenarios) is associated with a medium effect size and a lack of power explains why the difference does not remain significant after controlling for the number of comparisons. Overall, this suggests that personalized scenarios may be more anxiety provoking. Based on the multiple regression analysis, we can speculate this may be especially relevant for people who have a strong susceptibility to be immersed in VR. Nevertheless, the potential of using standardized scenarios in VR remains promising because it did elicit anxiety in GAD patients.

The findings are even more interesting because they were observed on the anxiety measure, but not on the less specific measure of negative affect. To be more precise, the impact of the exposure sessions mirror those of the anxiety measure on the a priori contrasts, but the differences do not remain statistically significant after controlling for the number of comparisons. Readers relying more on effect sizes than probability levels, or on power analyses, would consider the finding meaningful, consistent with Guitard et al. [25] and actually revealing more specificity to fear and anxiety than to diffuse negative emotions.

A pilot and independent clinical trial based on our results support our interest in the use of VR with GAD patients. Labbé, Thibault, Côté, and Gosselin [42] assessed the effectiveness of conducting only exposure to one standardized scenario (the emergency waiting room) in VR with people diagnosed with GAD. Participants were exposed three times to the scenario. Results showed a significant improvement on all measures related to GAD post-treatment, including the tendency to worry, symptoms of GAD, and anxiety. Treatment gains were maintained at the two-month follow-up. In addition, the changes were specific to health-related worries, which is consistent with the content of the scenario used for exposure. Results from Labbé et al. [42] are in line with the pioneering paper from Repetto et al. [43] on GAD, although they are the first to address the core fear of GAD.

Some limitations of the current study must be pointed out and discussed. First, the sample is relatively small. The provision of effect sizes should help gauge the magnitude of the experimental manipulations and plan larger studies. The effect of repeated exposure to standardized versus personalized scenarios should also be documented. The sociodemographic and clinical characteristics of the sample are typical of a study sample of GAD patients, with the exception of slightly more women than what is found in the general population, where women are usually three times more likely than men to have GAD [44]. A larger proportion of males would allow comparing the potential impact of gender differences. Documenting sex, economic, marital, and educational status in research articles is important for clinicians and researchers in order to appraise the sample and generalize the results. Reviews have been conducted on the power of immersions in VR to induce anxiety responses (e.g., [45]), but the impact of these variables has not yet been examined. Because these variables are frequently associated with anxiety disorders, their impact on the effect of VR deserve to be explored. The addition of physiological measures of anxiety would have documented and complemented our findings with objective measures [25]. However, heart rate or skin conductance would have been biased and unreliable given implicit differences in the exposure sessions. Participants were seated in the session of exposure in imagination. But in VR, participants were standing up and were physically moving when exploring the virtual environment. The intensity of the anxiety response also deserves attention. The research protocol was not designed to show how much an experience in VR could be frightening to GAD patients, but to show the potential of scenarios with a feeling of uncertainty to elicit anxiety in a population that is known to perceive uncertainty as threatening [18]. Finally, to increase generalization of the results to the psychotherapy contexts, it would have been interesting to conduct the study while patients are already in therapy and ready to proceed with exposure. Such a study comes with methodological challenges and it was considered better to first show that scenarios that are not individualized and presented in VR bear some potential.

Furthermore, the choice of three different VEs instead of only one could be argued as another limitation. However, the drawback of using only one scenario would be not targeting the main worry themes of the participants. This would be far more detrimental than comparing only three slightly different generic scenarios to 28 totally different and individualized ones. A replication study with a sample selected on the basis of the main worry theme would allow a more direct comparison of the exposure modalities with similar themes, or a larger sample would allow comparisons between virtual scenarios. Comparisons with people suffering from other anxiety disorders and with non-anxious participants would help document the specificity of the reactions to GAD.

The results from the exploratory analyses revealed that immersive tendencies, or individual predispositions to feel present, significantly predicted the increase in emotional reactions of participants. The predictive importance of the ITQ was significant when predicting anxiety and negative affect during exposure *in virtuo*. Perceived usefulness of worrying was another significant predictor of state anxiety in VR, but not intolerance of uncertainty. Presenting the neutral immersion in VR to all participants at the beginning of the experiment may have protected against the elements of novelty in the task [25], leaving room for other variables to stand out, such as the severity of dysfunctional thoughts about the usefulness of worry in predicting state anxiety. Future research should document with a larger sample, more predictors, better control for the different VR scenarios and planned hypotheses, and predictors of emotional reactions of patients in VR.

## 5. Conclusions

Because uncertainty is the core fear underlying GAD [18,19], the current study examined if immersion in virtual standardized scenarios that were developed based on the feeling of uncertainty and typical GAD worry themes may be relevant to be used in CBT. The increase in anxiety during immersion support the potential of VR for exposure, even in the case where feared stimuli are not as specific as in phobias and other anxiety disorders. This paves the way for the development of psychotherapy protocols that would integrate *in virtuo* exposure to test in randomized control trials.

## Figures and Tables

**Figure 1 jcm-08-00309-f001:**
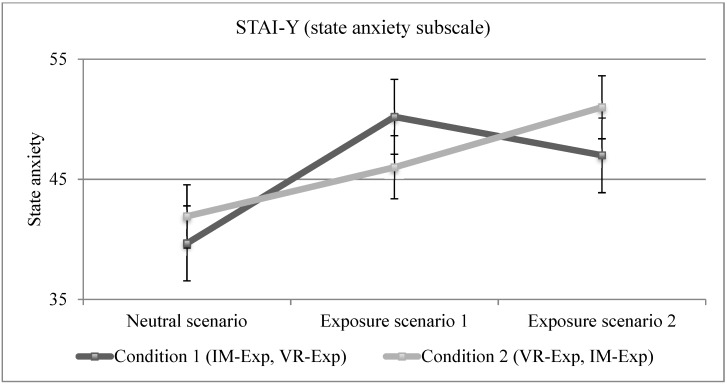
Illustration of the differential impact of exposure to a neutral scenario in virtual reality, a personalized scenario in imagination (IM-Exp) and a standardized scenario in virtual reality (VR-Exp) on the self-report measure of anxiety.

**Figure 2 jcm-08-00309-f002:**
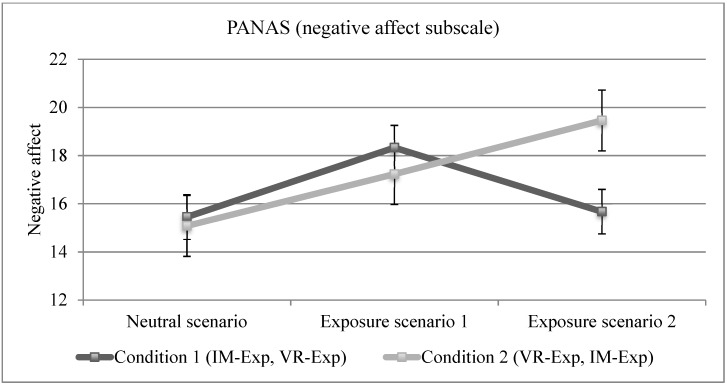
Illustration of the differential impact of exposure to a neutral scenario in virtual reality, a personalized scenario in imagination (IM-Exp) and a standardized scenario in virtual reality (VR-Exp) on the self-report measure of negative affect.

**Figure 3 jcm-08-00309-f003:**
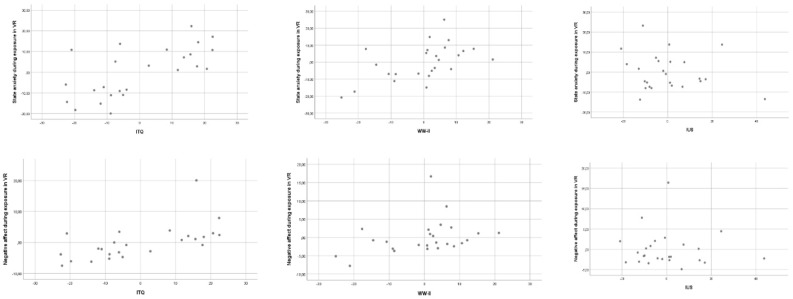
Scatterplots for the three predictors of state anxiety (**top** three) and negative affect (**bottom** three): immersive tendency (ITQ, **left** plot), positive beliefs about worry (WW-II, **center** plot), and intolerance of uncertainty (IUS, **right** plot).

**Table 1 jcm-08-00309-t001:** General description of the sample.

Variable	*N*	%	Mean (SD)
Nationality			
Canadian	25	89.3	
Senegalese	1	3.6	
Level of education			
University (some or completed)	18	64.3	
College or professional diploma	5	17.9	
High school diploma	3	10.7	
Some high school	2	7.1	
Socioeconomic status			
High	9	32.1	
Middle	15	53.6	
Low	4	14.3	
Marital status			
Married	10	35.7	
Single	8	28.6	
Common-law partner	8	28.6	
Divorced	2	7.1	
Descriptive clinical measures of generalized anxiety disorder			
Anxiety Disorders Interview Schedule-IV severity of GAD			5.7 (0.93)
Penn State Worry Questionnaire			59.88 (8.89)
Cognitive Avoidance Questionnaire			68.32 (20.35)
Descriptive measures of users’ experience in virtual reality			
Presence Questionnaire after the VR-Exp scenario			61.17 (19.05)
Gatineau Presence Questionnaire after the VR-Exp scenario			89.90 (14.32)
Simulator Sickness Questionnaire after the VR-Exp scenario			9.57 (6.65)
Predictive measures selected for exploratory analyses			
Intolerance of Uncertainty Scale			68.61 (21.62)
Why Worry-II Questionnaire			47.29 (17.93)
Immersive Tendencies Questionnaire			70.88 (16.81)

**Table 2 jcm-08-00309-t002:** Means and standard deviations of dependent variables in each experimental condition for the three scenarios.

Measure and Scenario	Condition			
	IM-Exp/VR-Exp		VR-Exp/IM-Exp	
	*M*	*(SD)*	*M*	*(SD)*
STAI-Y1				
Neutral environment	39.67	(10.83)	41.92	(13.12)
Exposure scenario 1	50.20	(12.81)	46.00	(12.91)
Exposure scenario 2	47.00	(12.66)	51.00	(14.74)
PANAS_NA				
Neutral environment	15.47	(5.14)	15.08	(4.25)
Exposure scenario 1	18.33	(6.23)	17.23	(5.12)
Exposure scenario 2	15.67	(3.60)	19.46	(8.61)

Note: IM-Exp = exposure to a personalized scenario in imagination, VR-Exp = exposure to a standardized scenario in virtual reality.

**Table 3 jcm-08-00309-t003:** Results of main effects of repeated measures ANOVAs for the comparative effect of cognitive exposure generalized anxiety disorder (GAD) scenarios presented in imagination and in virtual reality.

Effect	*MS*	*df*	*F*	*p*	*ηp* ^2^
STAI-Y1					
Time	564.10	2	9.03	< 0.001	0.258
Time × Condition	129.96	2	2.08	0.135	0.074
Condition	4.06	1	0.01	0.92	0.000
PANAS_NA					
Time	53.87	2	2.97	0.60	0.102
Time × Condition	48.75	2	2.69	0.078	0.094
Condition	56.38	1	0.61	0.44	0.023

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
