# Peer review of "Exposure to a Standardized Catastrophic Scenario in Virtual Reality or a Personalized Scenario in Imagination for Generalized Anxiety Disorder"

_jcm, 2019, doi:10.3390/jcm8030309_

Reviewer 1 Report

The proposed article is entitled "Exposure to a standardized catastrophic scenario in virtual reality or a personalized scenario in imagination. A comparison study for generalized anxiety disorder". It proposes, in the emerging field of cyberpsychology, to compare the anxious and affective responses of subjects confronted with a virtual reality device. The project is ambitious and its clinical interest is important in terms of management prospects.

The manuscript is generally well written and referenced. At first glance, the methodology seems rigorous and the statistical precautions seem to be taken in an adequate way. However, the number of subjects is questioned with regard to the method used or the objectives announced. 

The reader was surprised by the proliferation of questionnaires presented (14?) and was lost at first reading regarding the objectives of the study. Perhaps the form of the wording could be simplified or restructured so that the article becomes clearer.

The work is presented, via its title, as a comparative study. However, can we really talk about a comparative study with this limited number of subjects (28!). This is probably more than a preliminary study or preliminary results and therefore, perhaps modifying the title in this way would be appropriate.

Of these 28 subjects, we note a glaring disproportion between the number of men (4) and the number of women (24). However, sex is a discriminating variable concerning psychological disorders and in particular anxiety disorders. This point was addressed in the discussion but perhaps not sufficiently discussed. Comments regarding economic, marital or educational status in the discussion section would also have been relevant to highlight the limitations and recall that these variables are also frequently associated with anxiety disorders.

The authors present the approach used to remove outliers or, at least, neutralize their impact. While this approach is commendable, the question arises as to the number of outliers identified. Moreover, finding outliers on such limited data (28 subjects) raises questions. The procedure used is well described but is there not a risk of data smoothing around the average with such a procedure on such a small number of subjects?

The results part would benefit from being restructured into two sub-sections: one on the ANOVA model (description of the procedure and presentation of the results) and one on the regression model (description of the procedure and presentation of the results). Indeed, the results are dense and the reader may get lost in their reading. We suggest that lines 290 to 299 be placed just before lines 336 to 345. 

The statistical procedures used to compare groups are interesting... but perhaps very expensive in terms of statistical power. Has the number of subjects required to ensure results been calculated? This is all the more important as the authors seem to want to interpret the absence of a significant relationship as a result (304: "The interaction was non-significant, indicating that exposure to the traditional personalized scenario over time did not elicit more anxiety than exposure to the virtual scenario. »). It appears that the authors conducted two multiple regressions of STAI and PANAS scores with 7 predictors. If this is the case, it is likely that a larger number of subjects (about 50) will be required. The small number of subjects probably explains the difficulty in obtaining significant results.

In conclusion, the manuscript is interesting and its possible practical consequences are conceivable. However, in our opinion, some revisions and clarifications are still needed to improve it.

Author Response

We want to thank the reviewer for the thoughtful feedback and suggestions. All suggestions have been addressed and the text revised accordingly.

Please find the document attached.

Reviewer 2 Report

Thank you for the opportunity to review this study examining whether standardized worry scenarios in VR evoke negative emotions to the degree required for exposure therapy and comparable to imaginary exposure. I am in full agreement with the authors that worry exposure is the next frontier of VR therapy, and applaud the initiative. As to the study itself, I found it well-designed and executed. This will eventually make a fine contribution to the literature and hopefully inspire many similar studies in the future.

My main concern is that the current study is a follow-up study to a study not yet published (Guitard, Bouchard, Bélanger, “under review”). While I am sympathetic to the fact that circumstances may necessitate submission of two directly derived studies in parallel – and I am sure the authors have good reason to do it here – it puts me as a reviewer in somewhat of a disadvantage since this other study is not accessible to me. The current study is written as if this other study was published already and alludes to it many times, instead of being self-contained. This is not reason enough to reject the current study, but I cannot recommend publishing the current study before the other study has been published, at least not without substantial revisions.

 Other than that, I suggest only minor revisions:

* As the authors themselves discuss, power is likely an issue when contrasting reactions to imaginary and virtual scenarios. Please include a full post-hoc power analysis in the Methods section and reference it in the Discussion of limitations. In particular, it would be helpful to know what standardized mean difference that could be detected for this contrast with the current sample size (it would have had to be a large one).

* Some sort of visualization of the associations between predictors and change in anxiety/NA between neutral/exposure scenarios, would be helpful, preferably a set of scatterplots.

* Please add standard errors or confidence intervals to figures 1 and 2. Also, please adjust the y-axis: there is no need to cover the full possible score range and here it even obscures the relative difference which is of interest (i.e., zoom in by reducing the y axis range to better illustrate the differences between the two conditions/lines).

Author Response

We want to thank the reviewer for the thoughtful feedback and suggestions. All suggestions have been addressed and the text revised accordingly. You will find attached a revised version of the paper where the track change option was enabled to facilitate the reviewer’s work.

Round  2

Reviewer 1 Report

All suggestions were taken into account by the authors and the text was modified in a relevant way. We thank the authors for this article and hope that it will receive the scientific attention it deserves